# Gas Dynamics of Micro- and Nanofluidic Systems

Oleg Sazhin

**Abstract:** The size of micro- and nanofluidic devices accounts for their operation in modes that differ significantly from those for the corresponding macroscopic counterparts. Deep understanding of gas-dynamic processes occurring in micro- and nanofluidic systems opens new opportunities for the practical use of molecular transport at the micro- and nanoscale. Models and simulation methods with high reliability are described. The article also outlines the important flow parameters which must be considered in the first place to correctly simulate gas-dynamic processes in micro- and nanofluidic systems. The review will be useful as a reference for researchers interested in implementing preliminary analysis in the development and optimization of micro- and nanofluid devices.

**Keywords:** micro- and nanofluidics; Boltzmann equation; Knudsen number; simulation methods

## 1. Introduction

According to leading world experts, the industrial revolution in the first quarter of the 21st century was mainly associated with the progress of nanoengineering. Moreover, in terms of social impact, it is comparable with the revolution of the 20th century caused by the invention of a transistor, antibiotics, and IT. In the coming years, the development of science, engineering, and technology related to creation, research, and use of nanoscale objects will lead to fundamental changes in many fields of human activity, including material science, power industry, machinery manufacturing, electrical engineering, chemistry, medicine, agriculture, and ecology.

In recent years, technologies associated with micro- and nanofluidic engineering have been actively developed for a wide range of applications in various industries, including microelectronics, aerospace engineering, vacuum technology, nanoscale manufacturing, etc. [1–6]. Some examples of these devices are shown in Figure 1.

In terms of physical processes, there are no obstacles to miniaturization, the laws of thermodynamics are invariant with respect to sizes. However, currently, special attention is not paid to further nano-enabled miniaturization, but to new properties and effects that nanotechnologies can bring into the design of with micro- and nanoscale devices. To assess the actual level of the nano-industrial development, one can analyze the proceedings of one of the conferences related to the nanotechnological development, IEEE MEMS-2021 [7]. There, as many as 108 oral and 149 poster presentations were provided in 19 sections. Nanoelectromechanical systems (NEMSs) were addressed in 25 presentations (less than 10% of all reports). A significant proportion (about 50%) was accounted for by reports on microelectromechanical systems (MEMSs). Particular attention should be paid to the share of reports (about 30%) related to intelligent manufacturing technologies as well as to design and engineering innovations. The main result of further miniaturization is not the actual decrease in geometric sizes, but the improvement of quality characteristics based on new effects provided by small-size technology.

New tasks in the design and engineering of small-size technology are not only associated with the need of calculating and simulating problems related to circuitry and logic. There is also a necessity to solve a combination of problems related to engineering mechanics, heat and mass transfer, gas and hydrodynamics, etc., appearing separately or simultaneously in the product. Alternatively, carrying out exhaustive natural experiments

during the design and engineering of devices with micro- and nano-sized components is extremely difficult and economically unprofitable. In fact, it makes it necessary to develop the control and measuring equipment that is no less complex than the device itself (for example, for measuring the flows inside micro- and nanofluidic devices). Because the designing process in this case is often rather empirical, there is a need to develop, verify and apply new modeling methods in such systems.

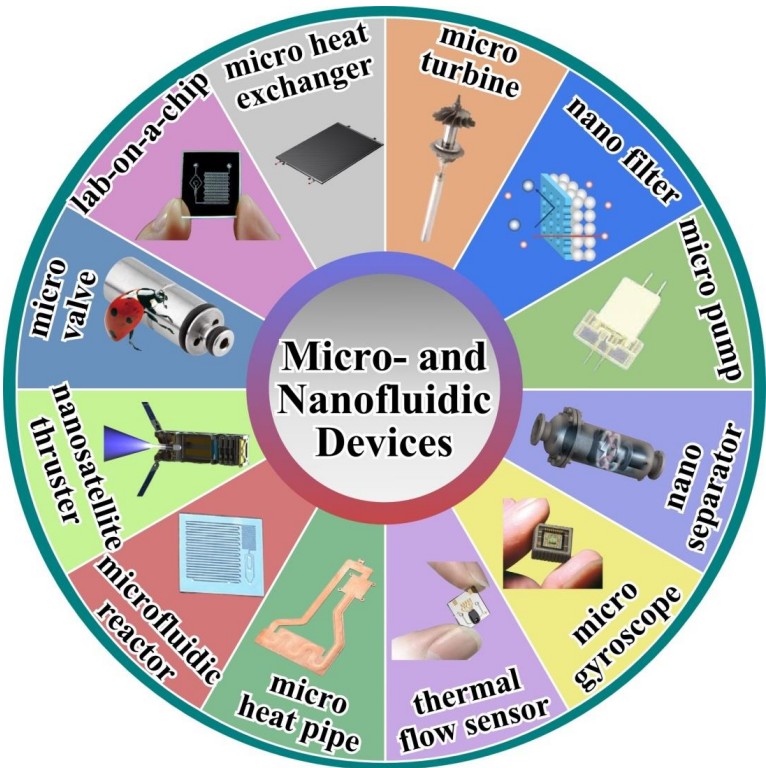

**Figure 1.** Micro- and nanofluidic devices.

Indeed, the size of micro- and nanofluidic devices accounts for their operation in modes that differ significantly from those for the corresponding macroscopic counterparts. An important effect of their small sizes lies in the fact that the traditional description of the gas flow in these devices as a continuous medium flow with macroscopic state parameters continuously varying in space and time becomes inadequate. This effect can be described with the use of the Knudsen number (Kn), which characterizes the gas rarefaction as the ratio of the mean free path of gas molecules to the characteristic linear size of the device. For example, under standard room air conditions, the Kn number approximates 0.07 for a typical device measuring 1 micron. Notably, for Kn numbers greater than 0.01, the assumption of the gas flow being a continuous medium is known to become invalid. Moreover, this effect increases substantially when the device is operated at high altitudes, as in aerospace applications, or at low pressures, as in vacuum technology, or has a size smaller than one micron.

## 2. Models and Simulation Methods

Classification of rarefied gas flow regimes based on the Knudsen number along with the applicable governing equations for each regime is shown in Figure 2. Initially, similar classification was proposed in Ref. [8]. Recall that, according to the Chapman–Enskog theory, the Euler equations (no viscosity and thermal conductivity in gas) represent a zero-order approximation of the Boltzmann equation; the Navier–Stokes and the Barnett equations represent a first-order and a second-order approximation, respectively. Physically, each subsequent approximation represents a new model of a continuous medium. As

follows from Figure 2, one possible way to correctly analyze the gas dynamics of micro- and nanofluidic systems in a wide range of gas rarefaction (Knudsen numbers) is to solve the Boltzmann integro-differential equation for the distribution function, which is the basic equation of the kinetic theory of gases.

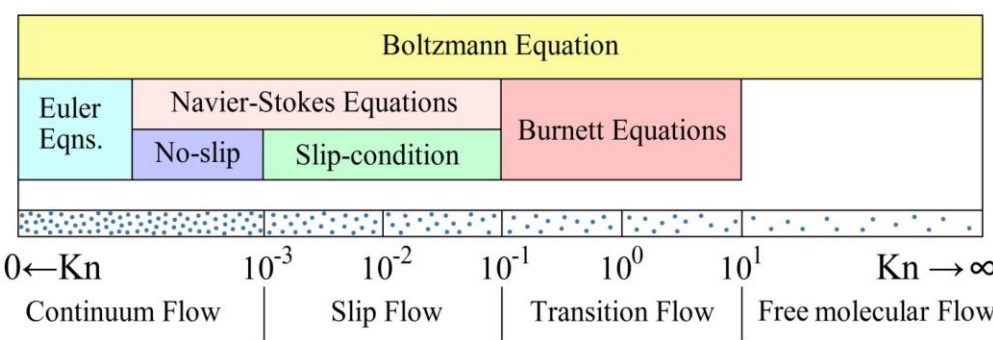

**Figure 2.** Classification of rarefied gas flow regimes based on the Knudsen number (Kn) and the applicable governing equations for each regime.

In the absence of external forces, the Boltzmann equation reads as

$$\frac{\partial f}{\partial t} + \mathbf{v} \cdot \frac{\partial f}{\partial \mathbf{r}} = Q(f, f_*), \tag{1}$$

where $f \equiv f(t, \mathbf{r}, \mathbf{v})$ is one-particle velocity distribution function, $t$ is time, $\mathbf{r}$ is a radius vector, and $\mathbf{v}$ is the molecular velocity vector. The distribution function is defined so as the quantity $f(t, \mathbf{r}, \mathbf{v})d\mathbf{r}d\mathbf{v}$ is the number of particles in the phase volume $d\mathbf{r}d\mathbf{v}$ near the point $(\mathbf{r}, \mathbf{v})$ at the time $t$. The right side of the Boltzmann equation is the collision integral.

$$Q(f, f_*) = \int K(\mathbf{v}', \mathbf{v}'_* \to \mathbf{v}, \mathbf{v}_*)(f'f'_* - ff_*)d\mathbf{v}'d\mathbf{v}'_*d\mathbf{v}_*. \tag{2}$$

The affixes to $f$ correspond to those of their arguments $\mathbf{v}$, in particular $f' \equiv f(t, \mathbf{r}, \mathbf{v}')$ and $f_* \equiv f(t, \mathbf{r}, \mathbf{v}_*)$. The quantity $K(\mathbf{v}', \mathbf{v}'_* \to \mathbf{v}, \mathbf{v}_*)$ is the probability density that two gas molecules having the velocities $\mathbf{v}'$ and $\mathbf{v}'_*$ after a collision have the velocities $\mathbf{v}$ and $\mathbf{v}_*$, respectively.

In essence, to determine function $K(\mathbf{v}', \mathbf{v}'_* \to \mathbf{v}, \mathbf{v}_*)$, we must solve a mechanical problem of two molecules colliding under to a given law. The most common and simplest law of intermolecular collision known so far is the hard sphere (HS) model. This is a special case $\eta = \infty$ of the repulsive inverse-power-law (IPL) potential $U(r) \sim r^{-\eta}$, where $r$ is the distance between molecules and $\eta$ is the power of IPL. According to this model, the molecular collision process is described by constant total cross-section $\sigma_t$ and isotropic scattering in the center-of-mass frame of reference. The HS model is suitable to simulate the elastic collision of gas molecules. The relationship between the coefficient of viscosity and temperature is postulated as $\mu \sim \sqrt{T}$.

A good fit to analytical calculations has the model of Maxwell molecules which represents a special case of the IPL potential where $\eta = 5$. Indeed, according to this model, the molecule collision probability is a constant value, which significantly simplifies both the theoretical analysis and the DSMC algorithm (the direct simulation Monte Carlo method).

The simplicity and ease of application are important virtues of both the HS and the Maxwell molecules models. However, none of them is applicable for simulating collisions in the case of real gas as they appear "too hard" and "too soft" models for the gas molecule–molecule interaction. The considerable disadvantage of these models is the absence of cross-section dependence on relative velocity of the colliding particles.

To eliminate this drawback, a model of variable hard spheres (VHS) has been developed [9]. According to the VHS model, which is defined for the general case of the IPL

potential, the diameter of colliding molecules $d$ is inversely proportional to the relative velocity of gas molecules $v_r$ to power $\omega - 1/2$, i.e., $\sigma_t \sim v_r^{-2\omega+1}$ where $\omega$ is a viscosity index derived from the temperature dependence of the viscosity coefficient. Indeed, both the IPL potential and the VHS model imply that the dependence of the viscosity coefficient $\mu$ on the temperature is proportional to $T^\omega$. Viscosity index $\omega$ is related to the power of IPL $\eta$ as $\omega = \frac{2}{(\eta-1)} + \frac{1}{2}$ and varies depending on the kind of gas.

The main drawback of the VHS model is that the theory based on this model leads to the formulas for viscosity $\sigma_\mu$ and diffusion cross sections $\sigma_D$ do not correspond to the chosen IPL potential. Indeed, according to the VHS model, the Schmidt number $(\sigma_\mu/\sigma_D)$ is constant and equals 3/2 as it does for the HS model. However, according to the IPL potential, this ratio should depend on $\eta$.

For the VHS model as for the HS model, the isotropic scattering law at molecular collision remains valid, but it does not appear realistic in general. Koura and Matsumoto suggested a variable soft spheres (VSS) model which can be used for any type of intermolecular potential with the definite viscosity and diffusion cross sections [10,11]. According to this model, diameters $d$ of the colliding particles depend on the relative velocity of molecules as in the VHS model, while the angle of the change in relative velocity direction (or deflection angle) $\chi$ is defined by the scattering law $b = d\cos^\alpha(\chi/2)$, which appears more realistic than the isotropic law (here $b$ is the impact parameter, and $\alpha$ is an exponent of cosine of deflection angle). The value of $\alpha$ should be chosen to coordinate values of diffusion and viscosity cross-section with the chosen interaction potential. Since in the case of the IPL potential the parameters involved in VSS model can be determined explicitly, the VSS model is used in practical calculations only for the IPL potential.

However, the IPL potential that considers only repulsion between molecules becomes incorrect in the low temperatures range where the attraction force becomes dominant. It is known that, under the condition $kT/\varepsilon >> 1$ the interaction potential can be considered to be fully repulsive, while in the case of $kT/\varepsilon << 1$ it is attractive; where $\varepsilon$ denotes well depth of the repulsive–attractive interaction potential of colliding particles, and $k$ is the Boltzmann constant.

In order to take attractive intermolecular forces into account, Hassan and Hash suggested to use the so-called generalized hard sphere (GHS) model of the intermolecular potential that contains both repulsive and attractive components [12]. In the GHS model, the post-collision direction of gas molecular relative velocity is the same as in the HS model, while total cross-section $\sigma_t$ is a function of the impact energy of the collision.

Kunc et al. promoted further development of the GHS model for the case of heavy gas molecules interacting through an attractive-repulsive potential where a strong attraction is assumed [13]. According to the authors, if intermolecular potential is represented as a sum of repulsive and attractive potentials $U = U^{att} + U^{rep}$, then specifically for the Lennard–Jones potential (LJ) with strong attraction $U^{att}/U^{rep} \sim (\varepsilon/E)^{1/6}$, where $E$ is impact energy of the collision. For instance, for Xe at the temperature of $T = 77.4$ K the ratio $(\varepsilon/E)^{1/6}$ is 1.15, at $T = 1000$ K $- 0.75$. Thus, attractive intermolecular forces can make a significant contribution into the gas intermolecular interaction in heavy gas even at rather high temperatures.

A generalized soft sphere (GSS) model introduced by Fan implies that the total cross-section is the same as in the GHS model while the deflection angle similar to that in the VSS model [14]. In the case of the LJ potential, a simplifying assumption was employed by fitting the formula for collision integrals, which allowed to determinate explicitly parameters involved in the GSS model.

Matsumoto developed a variable sphere (VS) model that provides consistency viscosity $\sigma_\mu$ and diffusion cross sections $\sigma_D$ with those of any realistic intermolecular potential and obeys a simple scattering law at collision that is even simpler than isotropic one [15]; however, even the author himself noted that the model should be examined with many rarefied gas flow problems.

To solve the problems of external and internal gas flow around the surface of a solid body based on the Boltzmann equation, it is necessary to correctly state the boundary

conditions. The problem is to express the distribution function of molecules scattered by the surface as $f^+ \equiv f(t, \mathbf{r}, \mathbf{v})$ $(v_n = \mathbf{v} \cdot \mathbf{n} > 0)$ through the distribution function of molecules incident on the surface $f^- \equiv f(t, \mathbf{r}, \mathbf{v}')$ $(v_n' = \mathbf{v}' \cdot \mathbf{n} < 0)$, where $\mathbf{n}$ is the unit vector normal to the surface, $\mathbf{v}'$ and $\mathbf{v}$ are velocities of incident and reflected molecules, respectively.

There is a generally accepted theoretical method used to describe the scattering of gas molecules by a surface, which is based on the construction of mathematical scattering kernels [16]. The essence of this method is to calculate the probability density $R(\mathbf{v}' \rightarrow \mathbf{v})$ that a molecule would hit the surface with a velocity between $\mathbf{v}'$ and $\mathbf{v}' + d\mathbf{v}'$ at point $\mathbf{r}$ at time $t$ flies out of the same surface element at the same time at a velocity between $\mathbf{v}$ and $\mathbf{v} + d\mathbf{v}$. Then, in an integral form the boundary conditions for the Boltzmann equations can be written as follows

$$|v_n| f^+ = \int_{v_n' < 0} |v_n'| R(\mathbf{v}' \rightarrow \mathbf{v}) f^- d\mathbf{v}'. \tag{3}$$

The mathematical notation of function $R(\mathbf{v}' \rightarrow \mathbf{v})$ is called the scattering kernel. It includes all information about the influence of physical and chemical properties of the gas–solid interface on the gas–surface scattering process.

In order to obtain a scattering kernel, an appropriate physical model of the gas–surface scattering process had to be developed. It was Maxwell who introduced it for the first time [17]. According to this model, fraction $\alpha_M$ of the incident molecules comes to equilibrium with the surface and reflects from it diffusely, while the remaining part $(1 - \alpha_M)$ is reflected specularly. The Maxwell model has the following scattering kernel

$$R_M(\mathbf{v}' \rightarrow \mathbf{v}; \alpha_M) = (1 - \alpha_M) \cdot \delta\big(\mathbf{v} - (\mathbf{v}' - 2v_n'\mathbf{n})\big) + \alpha_M \frac{m^2 v_n}{2\pi (kT_s)^2} \exp\left(-\frac{mv^2}{2kT_s}\right), \tag{4}$$

where $m$ is the molecule mass, $T_s$ is the surface temperature, and $\delta(x)$ is the delta function. Obviously, the Maxwell's scattering kernel is based on the relatively simple physical model and thus this scattering kernel has a number of inaccuracies when it comes to describing the gas scattering by the surface. In particular, one parameter $\alpha_M$ characterizes the exchange in both energy and momentum of the gas molecules with the surface. However, it is known that the transfer rates of energy and momentum during gas–surface collisions are different from each other, and usually the momentum is lost or acquired more rapidly than the energy. This is one of the shortcomings of the Maxwell's scattering kernel.

The Cercignani–Lampis (CL) [18] and Epstein [19] scattering kernels are more physically sound. In particular, they make it possible to derive the dependence of the gas–surface scattering process from the ratio between the velocity of gas molecules incident on the surface and the surface temperature. This makes a fundamental distinction from the Maxwell's scattering kernel. Indeed, since $\alpha_M$ is a constant, the outcome of the gas–surface scattering (diffuse or specular) does not depend on this ratio. This fact is important for studying non-isothermal gas flow. To prove that, we were able to show that the simulation based on CL and Epstein scattering kernels provide a satisfactory description of the gas–surface scattering under a non-isothermal rarefied gas flow than the commonly used Maxwell's kernel [20].

In the case of isotropic gas–surface scattering in the plane tangential to the surface, the mathematical form of the Cercignani–Lampis scattering kernel can be written as

$$R_{CL}(\mathbf{v}' \rightarrow \mathbf{v}; \alpha_n; \alpha_\tau) =$$
$$= \frac{m^2 v_n}{2\pi \alpha_n \alpha_\tau (2 - \alpha_\tau)(kT_s)^2} \exp\left\{ -\frac{m\left(v_n^2 + (1 - \alpha_n)v_n'^2\right)}{2\alpha_n kT_s} - \frac{m(\mathbf{v}_\tau - (1 - \alpha_\tau)\mathbf{v}_\tau')^2}{2\alpha_\tau (2 - \alpha_\tau)kT_s} \right\} \times I_0\left( \frac{\sqrt{1 - \alpha_n}}{\alpha_n} \frac{mv_n v_n'}{kT_s} \right); \tag{5}$$
$$(0 \leq \alpha_n \leq 1; \ 0 \leq \alpha_\tau \leq 2),$$

where $I_0(x) = (2\pi)^{-1} \int_0^{2\pi} \exp(x \cos \varphi) d\varphi$, $\mathbf{v}_\tau$ is the two-dimensional vector of the tangential velocity, $\alpha_n$ is the accommodation coefficient of the kinetic energy corresponding to the normal molecular velocity, and $\alpha_\tau$ is the tangential momentum accommodation coefficient.

The Epstein scattering kernel is a generalization of the Maxwell one. Epstein suggested that the kernel parameter $\alpha_M$ depends on the velocity of gas molecule incident on the surface and obtained the following scattering kernel

$$R_{Ep}(\mathbf{v}' \to \mathbf{v}; \alpha_M(\mathbf{v}')) =$$
$$= (1 - \alpha_M(\mathbf{v}')) \cdot \delta(\mathbf{v} - (\mathbf{v}' - 2v'_n \mathbf{n})) + \frac{v_n \exp(-mv^2/2kT_s)\alpha_M(\mathbf{v})\alpha_M(\mathbf{v}')}{\int_{v_n > 0} v_n \exp(-mv^2/2kT_s)\alpha_M(\mathbf{v})d\mathbf{v}}, \tag{6}$$

where $\alpha_M(\mathbf{v}')$ is a function that qualitatively describes the temperature dependence of the energy accommodation coefficient, and wrote it as

$$\alpha_M(\mathbf{v}') = \exp\left(-\frac{mv'^2}{2k\theta_1}\right) + C\left[1 - \exp\left(-\frac{mv'^2}{2k\theta_2}\right)\right]. \tag{7}$$

This function includes three parameters: $\theta_1$, $\theta_2$, and $C$, where $\theta_1$ and $\theta_2$ control the behavior of $\alpha_M$ for low and high velocities of the incident molecules, and $C$ is the limit value $\alpha_M$ in the region of high velocities of the incident molecules. Usually, the Epstein kernel parameters are determined for each specific gas–surface system to provide an acceptable agreement between analytical expression obtained with the use of this scattering kernel and appropriate experimental data. Obviously, expression (7) is not the only possible function $\alpha_M$. The necessary conditions for $\alpha_M$ include its evenness and $0 \le \alpha_M \le 1$. Another form of $\alpha_M$ was proposed in [21] where it is provided as a function of the kinetic energy of the motion normal to the surface.

The solution of the Boltzmann equation is a rather complicated computational problem. Therefore, when using the kinetic approach to solve an array of problems related to the description of rarefied gas flows, the collision integral is often replaced by a simpler approximate expression that preserves the main properties of the exact collision operator [16]. The resulting equation is called the model kinetic equation or the Boltzmann equation model. Undoubtedly, the most common model of the Boltzmann equation is the BGK model [22], which was developed in the ES and S models [23,24]. The latter two models, unlike BGK, predict the correct Prandtl number upon transition to the hydrodynamic regime. The subsequent development of Boltzmann equation models aimed to solve specific problems, in particular, for rarefied flows of polyatomic gases [25–27] and gas mixtures [28].

To date, a sufficient number of methods for solving the exact Boltzmann equation and its models have been developed and successfully applied in practice. Classification of these methods is due to the diversity of problems to be solved, including restrictions, dimensions, approximations, assumptions, and other requirements, not to mention the own preferences of the classification author. Indeed, it is not infrequent that a certain method shows excellent results (in good agreement with the experiment) for one class of problems but appears to be unsatisfactory or totally unacceptable for another. For example, the methods of solving the Boltzmann linearized equation models have been tried and verified in calculating the rarefied gas flow in long channels, where the flow is fully developed. However, these methods become obviously unapplicable when the flow becomes strongly non-equilibrium, such as in the case of the gas outflow into a vacuum through slits and short channels.

Noteworthy is the classification by G. A. Bird, who divided the methods into analytical and numerical as applied to the transition regime of the rarified gas flow [29]. The first category includes methods for solving moment and model kinetic equations, most of which require the involvement of numerical procedures to achieve the final result. The numerical category is further divided into deterministic and stochastic (probabilistic) simulations, direct Boltzmann computational fluid dynamics (CFDs) and discretization methods.

In our opinion, if we adhere to a rougher classification without going into details of the method, then it is appropriate to simply divide the methods into deterministic and stochastic. The most common deterministic method for solving the Boltzmann equation is the discrete velocity method proposed by J. E. Broadwell [30]. The essence is to construct a

discrete analogue of the original equation in the velocity space. For this purpose, a velocity grid is introduced as fixed set of available gas molecular velocities, and the collision integral is approximated at the grid nodes of a specially fitted quadrature formula. With this approach, the determination of the distribution function values at the grid nodes is reduced to solving a system of ordinary differential equations, in which the molecular velocities are included as the parameters.

In recent years, the problems related to fluid simulation have been widely approached by the lattice Boltzmann method (LBM). First introduced by McNamara and Zanetti [31], this method is based on the numerical solution of the Boltzmann equation, which is discrete in time, space, and velocity. Notably, the LBM shows stable results when the fluid velocity is much lower than the sound velocity in the medium, i.e., the lower the velocity, the better the result [32]. In particular, the LBM version proposed in [33] is applicable at Mach numbers M < 0.15.

As deterministic, G.A. Bird also classified the molecular dynamics (MDs) method, in which the temporal evolution of the interacting particle system is tracked by integrating their motion equations [34]. It should be noted that the MD method is not widely used for calculations of rarefied gas flows. In our opinion, it is advisable to apply this method in a hybrid approach, in which the MD method is "responsible" for modeling the interaction of gas molecules with the surface.

Stochastic approaches to solving the Boltzmann equation began to develop actively with the development of computer technology. In particular, an algorithm was developed for solving a system of stochastic equations in the Poisson measure, which represents the approximate solution of the Boltzmann equation [35]; the Galerkin method [36]; the collocation method [37]; the weighted particle method [38]; etc. However, the most popular stochastic approach to solving the Boltzmann equation is, by far, the direct simulation Monte Carlo method (DSMC) [29]. This is an effective tool for solving problems of rarefied gas dynamics for any gas flow regime from a continuum to a free molecular flow.

Recall that the DSMC method simulates gas flows using model particles which represent a large number of real molecules: at each discrete time interval, particle motion and intermolecular collisions are considered (simulated) as two independent and unrelated processes. If the time interval is less than the mean free path between collisions, then the method provides physically realistic results. The method allows taking into account many factors, such as strong non-equilibrium and complex geometric configuration of the system being modeled, using various types of boundary conditions (gas–surface scattering laws), surface structure, and intermolecular potential models.

We have experience in successfully applying the DSMC method to investigate gas-dynamic processes in specific configurations of micro- and nanofluidic systems, as seen in our recent works [39–41]. In our studies, we used our own program codes implementing the DSMC method based on the majorant frequency scheme [42]. We also made use of non-uniform meshes and special DSMC procedures such as cell division into sub-cells and weight planes. As a result, we normally received precise values of dependencies of the mass flow rate and the flow field through channels of various shapes and configurations from a broad set of defining parameters: gas rarefaction (Knudsen number); reduced length (length to channel height ratio); pressure ratio (output–input); gas–surface scattering and gas molecule–molecule interaction parameters. All calculations were performed using multiprocessor computer hardware. As an example of these calculations, Figure 3 shows a fragment of the flow field inside and downstream a sudden expansion channel in the near continuum flow regime [39]. Indeed, in practice the configurations of micro- and nanofluidic devices often represent channels with complex geometry, particularly those with sudden expansion or contraction. As can be seen from the figure, the gas flow in micro- and nanofluidic devices can have a rather complex structure and requires a more detailed study.

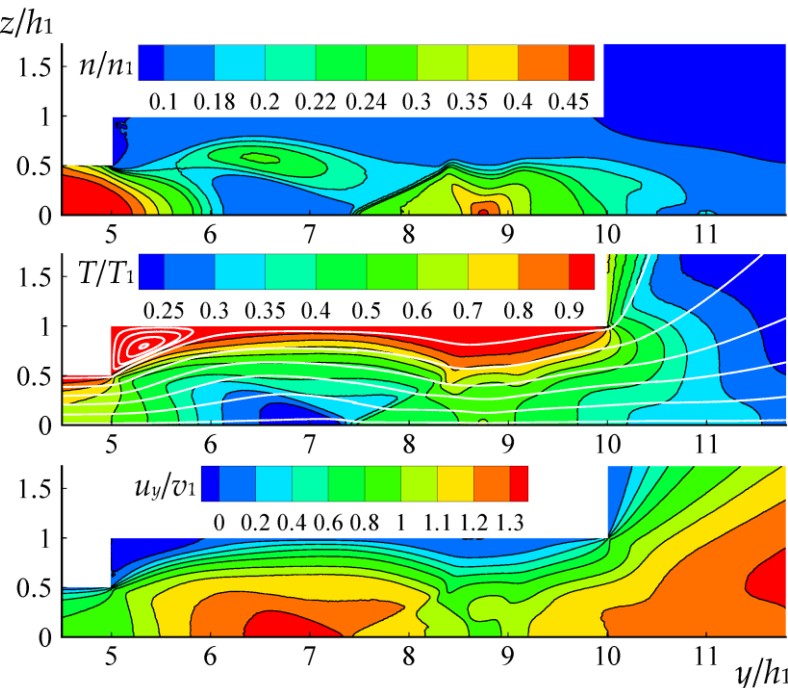

**Figure 3.** Dimensionless density $n/n_1$, temperature $T/T_1$, and longitudinal bulk velocity $u_y/v_1$ in the yz-plane inside and downstream a channel ($l/h_1 = 10$) with sudden expansion ($h_2/h_1 = 2$) at rarefaction $\delta = 10^3$ (the temperature image also displays gas streamlines) [39].

It should be noted, however, that the DSMC method, due to its stochastic nature, has a drawback of high statistical noise in modeling low-speed flows. In this case, it is advisable to use deterministic methods, which are deprived of this shortcoming. In particular, the deterministic discontinuous Galerkin fast spectral (DGFS) method was recently developed for solving the exact Boltzmann equation [43,44]. It allows arbitrary unstructured geometries; high order accuracy in physical space time and velocity space; and arbitrary collision models and provides excellent nearly linear scaling characteristics on massively parallel architectures. It was shown that the DGFS method reproduces noise-free smooth results of assessing the flow phenomenon in the thermostress convection processes in the slip-to-transition flow regime [45]. If we adhere to a detailed classification of methods, then the spectral methods for solving the Boltzmann equation can also be classified as a separate category. A detailed classification of methods for solving the Boltzmann equation can be found in [16,46,47], and specifically deterministic approaches are reviewed in [48–50].

### 3. Important Flow Parameters

In addition to correctly selecting the method, one should consider that devices comprising micro- and nanocomponents exhibit the phenomena associated with the accommodation properties of gas molecules and the surface structure. Indeed, the actual interaction of gas flows with the surface of micro- and nanofluidic devices can differ significantly from the assumption which is used in most engineering calculations, that there is a complete accommodation of the energy and momentum of gas molecules on the surface, with corresponding accommodation coefficients assumed to be equal to 1. In particular, we experimentally showed that the tangential momentum accommodation coefficient on an atomically clean surface can reach 0.7, which increases the gas flow rate through the channel used in the experiment by 65% compared with the theoretical value under the assumption of complete accommodation of the tangential momentum [51]. The effects associated with incomplete energy accommodation can be even more significant: depending on the chemical composition of the surface, the energy accommodation coefficient can drop dramatically

to as low as the order of two decimal places [52]. This, without a doubt, can greatly affect the heat transfer in the device being used.

In some cases, the geometric sizes of the operating elements are such that the structure of the surface and the interface plays a decisive role in the operation of the respective device. For example, the surface in micro- and nanofluidic devices can vary from quite smooth, as in the case of a crystalline silicone surface, to rather rough, as with the technical surface. The role of surface roughness in a rarefied gas flow inside a channel was demonstrated in [53–55].

In addition, such a parameter as a surface-to-volume ratio has a tangible effect on heat transfer processes. Obviously, it is significantly different for micro- and macrodevices.

Thus, the following flow parameters must be taken into account to correctly simulate gas-dynamic processes in micro- and nanofluidic systems:

- Gas rarefaction (Knudsen number);
- Gas velocity as a whole (Mach number or pressure gradient);
- Temperature non-equilibrium (temperature ratio);
- Gas–surface scattering (scattering kernel parameters or accommodation coefficients);
- Gas molecule–molecule interaction (intermolecular potential parameters);
- Surface roughness (surface structure model parameters);
- System geometry (shape, length-to-height ratio, aspect ratio);
- Surface-to-volume ratio.

Possibly, and very likely, the above list is incomplete, and so can be supplemented by the reader. However, we brought together the important flow parameters which must be considered in the first place, in particular to implement preliminary analysis in the development and optimization of micro- and nanofluid devises, for example, in space application design and vacuum engineering.

## 4. Conclusions

We attempted to review an extensive topic of gas dynamics in micro- and nanofluidic systems. The article discusses models and simulation methods based on the Boltzmann integro-differential equation for the distribution function. The flow parameters presented here can be used to correctly simulate gas-dynamic processes in micro- and nanoscale.

We believe that a deep insight into the gas-dynamic processes occurring in micro- and nanofluidic systems opens up new opportunities for the practical use of molecular transport at the nano- and microscale. In particular, nano- and microscale technologies make it possible to achieve a specific output that is unreachable by "macrotechnologies" (as for microturbines and micropumps); to improve the efficiency of gas mixture separation with the use of micro- and nanoseparators (which is of key importance for nuclear power production); to design small-size and high-precision micro total analysis systems (or lab-on-a-chip); to enhance the performance of microgyroscopes, microsensors, micron heat pipes, etc.

**Funding:** This research was supported by the Russian Science Foundation and Government of Sverdlovsk region, Joint Grant No. 22-21-20121, https://rscf.ru/en/project/22-21-20121/.

**Conflicts of Interest:** The author declares no conflict of interest.

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
