# Peer review of "Gas Dynamics of Micro- and Nanofluidic Systems"

_fluids, doi:10.3390/fluids8010024_

Round 1

Reviewer 1 Report (Previous Reviewer 2)

The resubmitted paper is improved, but still needs some revisions before publication.
the introduction is to be extended.

The paper still relatively short.
A bibliometric analysis is to be performed.
More papers related to the subject are to be classified and discussed.

The cited references are to be described with more details.

Author Response

See attachment, please. 

Reviewer 2 Report (New Reviewer)

The manuscript deals with the recent advances in fluid dynamics for micro- and nanofluidic systems. Authors describe modern models and simulation methods used in nanofluidic science and flow parameters important for such systems.

Reviewer's comments and suggestions for Authors

1) Abstract is seemed to be too short

2) I suggest to delete the sentence “The current economic, military, social and political condition of developed countries is mainly determined by the development of new IT and national nanoindustry” (Line 15). Because “Fluids” is not about “economic, military, social and political condition”.

3) English should be checked and improved. For example

- The sentence starting at line 16 should be in Present Time. (is mainly associated)

- In line 20 the expression “all combined” is redundant (or the sentence should be rewritten)

- I do not like “From there,” in line 40 (though it is grammatically correct)

and so on

4) Figure 1 needs more explanation and description.

5) Some parts of the text have no reference. For example, the paragraph between 57 and 69 lines. This is an incorrect manner for a review paper. In the same time, the total number of reference is rather low.

6) Figure 3 is not displayed correctly.

7) According to the Abstract, “the important flow parameters” are one of the main topic of the manuscript. However, this section is less than one page.

Author Response

See attachment, please.

Round 2

Reviewer 2 Report (New Reviewer)

The author has substantially improved the manuscript. In my opinion, it can be published in the present form.

This manuscript is a resubmission of an earlier submission. The following is a list of the peer review reports and author responses from that submission.

Round 1

Reviewer 1 Report

The author has written a short bibliographic review about “gas dynamic of micro- and nanofluidic systems”. His review offers several advices concerning the calculation to predict the molecular transport in such small channels. He clearly underlines the factors that should be considered for a proper simulation and the main adapted modeling techniques as a function of Knudsen number.

Minor concerns:

-figure 2: did the authors provide all copyrights related the small photographies or schemes.

-Figure 2 and introduction, considering lab-on-a-chip for modeling molecular transport in gas phase seems to be strange since more than 95% of lab-on-a-chip are using liquid phase.

-The text after figure 2 is highlight with a yellow color, please remove this colored text.

-Please check that all abbreviations are clearly given in the text

-Figure 3, the author should give more explanation how did he carry out such simulations.

-There is many self-citations in this review and the author should add more citations of other colleague to improve the end of the review

My general feeling after reviewing this article, is that the authors did a nice work and this article should be published after minor revisions.

Reviewer 2 Report

the author presented a very short review paper on Gas dynamics of micro- and nanofluidic systems.

The paper can not be accepted in the present form and should be extensively improved before resubmission (the author must refer to some previously published review papers to get an idea on what a review paper should contain).

The paper doesn’t contain any section or sub-section.

More papers related to the subject are to be classified and discussed.

The cited references are to be described with more details.